# Awareness and Attitude of Physicians on the Role of Dentists in the Management of Obstructive Sleep Apnea

**DOI:** 10.3390/ijerph192316126

**Published:** 2022-12-02

**Authors:** Najla S. Alrejaye, Khalid M. Alfayez, Hafsah H. Al Ali, Yara M. Bushnaq, Reem S. Bin Zaid, Fahad K. Alobaid, Mostafa A. Abolfotouh

**Affiliations:** 1King Abdulaziz Medical City, Ministry of National Guard Health Affairs, Riyadh 11426, Saudi Arabia; 2College of Dentistry, King Saud bin Abdulaziz University for Health Sciences, Riyadh 14611, Saudi Arabia; 3King Abdullah International Medical Research Center, Riyadh 11481, Saudi Arabia; 4College of Dentistry, Riyadh Elm University, Riyadh 13244, Saudi Arabia; 5College of Dentistry, King Saud University, Riyadh 11451, Saudi Arabia

**Keywords:** attitude, sleep-related breathing disorder, Saudi

## Abstract

Background: Efforts to evaluate physicians’ awareness and attitude toward dentists’ comprehensive role in OSA management are relatively negligible. Therefore, this study aimed to assess physicians’ awareness and attitude toward the role of dentists in OSA management in Saudi Arabia. Methods: In a multi-center cross-sectional study, a total of 358 physicians in Saudi Arabia were subjected to an e-questionnaire composed of three sections: (1) physicians’ demographic data, (2) physicians’ general and specific knowledge of OSA and its management [using 29 factual statements to be responded by “True, False, or I don’t know” responses], and (3) physicians’ attitude towards dentists’ role in OSA management [using 12 attitude statements to be responded by a Likert scale of “Never, Rarely, Sometimes, Usually, Always” responses]. A scoring system was applied for both knowledge and attitude, total and percentage mean scores (PMS) were calculated, and knowledge and attitude levels were categorized accordingly. Predictors of correct knowledge and favorable attitude were identified using multiple regression analyses. Results: Physicians had an overall average knowledge level (PMS = 56% ± 19.4%), with 35.5% and 5.9% reporting good general and specific knowledge levels, respectively (χ^2^ = 143.0, *p* < 0.001). Physicians had an overall neutral attitude level (PMS = 64.4% ± 17.5%), with about one-half reporting a neutral attitude level (48.9%) and only one-fourth reporting a positive attitude level (27.7%). Higher levels of knowledge were a significant predictor of favorable attitudes (*t* = 5.71, *p* < 0.001). Higher training levels were a significant predictor of correct knowledge (*t* = 3.60, *p* < 0.001) and favorable attitude (*t* = 3.15, *p* = 0.002). Conclusions: Physicians showed insufficient knowledge about OSA and a less than favorable attitude towards dentists’ role in its management. Enhancing medical curricula and clinical protocols and guidelines on the dentists’ role in OSA management is recommended.

## 1. Introduction

Obstructive sleep apnea (OSA) is a common breathing disorder in which recurrent episodes of upper airway obstruction occur during sleep. The main symptoms include nocturnal respiratory pauses interrupted by loud intermittent snoring and excessive daytime sleepiness [1]. The prevalence of OSA is relatively high world-wide. In North America, OSA prevalence was found to be approximately 15–30% in men and 10–15% in women in North America [1,2]. The prevalence of moderate to severe OSA in Switzerland was reported to be 23.4% in women and 49.7% in men [3]. OSA prevalence seems to be also high among Saudi adult population. Based on studies that used the Berlin questionnaire to screen for OSA symptoms, high-risk individuals for OSA were present in three out of ten middle-aged Saudi males and in four out of ten middle-aged Saudi females [4,5], whereas the prevalence among Saudi population was about 12.8% in men and 5.1% in women, based on a study used the Wisconsin questionnaire [6]. Unfortunately, the vast majority of OSA remain undiagnosed and untreated [7], although OSA may have serious complications affecting cardiovascular system if left unmanaged [8].

The predisposing factors of OSA vary. It could be mainly due to craniofacial discrepancies (micrognathia and retrognathia) in non-obese individuals rather than a soft tissue related factor from upper airway soft tissue enlargement, which is mainly found in obese individuals [9,10]. Dentofacial deformities and abnormal craniofacial development may result in breathing disorders such as mouth breathing and OSA [11]. Accurate diagnosis and efficient management of OSA may substantially improve quality of life. Studies showed that dentists may have a significant role in OSA management. For instance, maxillary orthopedic expansion or distraction may help in managing OSA in children [12,13]. Moreover, maxillomandibular (double jaw) advancement may significantly reduce apnea-hypopnea Index (AHI) in adults [14]. Moreover, mandibular advancement oral appliances were reported to have similar effectiveness to positive air pressure (PAP) therapy due to superior patient preference and adherence [15]. It is important to mention that patient selection is a key for successful management and PAP therapy remains the gold standard for OSA management. Polysomnography is essential for OSA diagnosis, which should be assessed by a sleep physician [16].

OSA related to a craniofacial or dentofacial deformity may be best managed by a collaboration between physicians and dental specialists. Timely referral of such OSA cases to dentists, may facilitate timely intervention, reduce serious systemic complications, and have a better functional and esthetic results [11]. Interestingly, Koufatzidou et al. found that only one-third of pediatricians in Greece examined or referred their patients for dentofacial problems such as crossbite and overbite, although, the role of physicians is crucial in actively acknowledging and referring those cases of malocclusion to an orthodontist in a timely manner [17]. Furthermore, Sri Meenakshi et al. conducted a survey to assess the role of medical and dental practitioners in OSA diagnosis, its management, and their referral frequency to orthodontists [18]. They found that the awareness regarding the diagnostic options, management, and consequences of untreated OSA was insufficient. Moreover, the knowledge of both groups about orthodontists’ role was poor and referral to orthodontics for OSA management was consequently rare [18]. Nonetheless, the scope of the study was confined to orthodontics only. Studies in the literature that assessed perception of physicians towards the comprehensive role of dentists in OSA management are limited. Thus, the aim of our study was to assess the level of awareness and attitude of physicians who commonly encounter patients with OSA (primary care physicians, pediatricians, ENT specialists, pulmonologists and sleep physicians) towards the role of dentists in the management of obstructive breathing disorders, mainly OSA, in Saudi Arabia. This study may aid in emphasizing the importance of raising the awareness to help in timely referral, proper diagnosis, and efficient management of patients with OSA.

## 2. Materials and Methods

### 2.1. Study Participants

Study participants included primary care physicians (general practitioners, family physicians, and internists), pediatricians, ENT specialists, pulmonologists, and sleep physicians working in different regions in Saudi Arabia.

### 2.2. Study Design

A multi-center cross-sectional study design was used. 

### 2.3. Study Population and Sampling Technique

The questionnaire was distributed electronically between August 2021 and November 2021, targeting the participants of interest in different regions in Saudi Arabia (Northern, Central, Western, Eastern, and Southern) to ensure good representation from across the country. Proportional quota sampling was used to ensure that respondents were demographically representative of the general population with quotas based on region and specialty. Based on the assumption of 50% level of positive perception towards the role of dentists in the management of OSA, and with a confidence interval of 95% and a margin of error of 5%, a sample size of 384 physicians was estimated. Those who responded with valid completed questionnaires were 358 physicians, after excluding responses from dentists and other non-targeted specialties.

### 2.4. Data Collection Methods

Based on the literature [18,19], with some modifications, an e-questionnaire was created and validated by the expert opinion. The questionnaire is composed of three sections: (1)Demographic data such as: gender, age, specialty, and years of experience;(2)Knowledge about OSA and the role of dentists. Using 29 factual statements to be responded by “True, False, or I don’t know” responses, data was collected about the physicians’ knowledge on the followings: (a) *General knowledge* of craniofacial factors that may cause or aggravate obstructive breathing disorders and general OSA management (statements # 1 to 12), and (b) *Specific knowledge* about the role of different dental specialties in the management of OSA: orthodontics, oromaxillofacial surgery, and general dentistry (statements # 13 to 29). A scoring system was applied to assess the level of knowledge of each subject: 1 point was given for each correct answer, and 0 points were given for each incorrect or an ‘I don’t know’ answer. Total and percentage mean scores (PMS) were calculated. Participants were grouped into three categories according to their levels of knowledge: poor (<50% PMS), average (50–75% PMS), and good (>75% PMS);(3)Attitude towards the role of dentists in OSA. Using 12 attitude statements to be responded by a Likert scale of “Never, Rarely, Sometimes, Usually, Always” responses, data was collected on the attitude towards the recognition of craniofacial deformities that may contribute to OSA and their referral response. A scoring system was applied using the Likert 5-point scale; 5 points were assigned to “Always,” and 1 point was assigned to “Never.” Negative attitude statements were scored from 1 (for those who selected always) to 5 (for those who selected never). Total attitude score and PMS were calculated, and participants were grouped into three categories according to their PMS as follows: positive attitude (>75% PMS), neutral attitude (50–75% PMS), and negative attitude (<50% PMS).

A pilot study was conducted on ten subjects to test the reliability and feasibility of the questionnaire. The questionnaire was assessed in terms of internal consistency. Cronbach’s alpha was computed, and a coefficient alpha of 0.83 was considered adequate. Test-retest reliability was also assessed using Cronbach’s alpha and Pearson’s correlation coefficient (r). Construct validity of the checklist was assessed using expert opinion, and the final version was approved after making the necessary modifications.

### 2.5. Data Analysis

SPSS software Ver. 27 was used for data entry and analysis. Descriptive statistics such as mean score, standard deviation, frequency and percentages of all independent variables (age, gender, educational grade, etc.) were used. Responses were scored by frequency and percentage, converted to percentage mean scores, then transformed into qualitative data. For qualitative data, Pearson’s chi-squared test, the chi-squared test for linear trends, and McNemar’s test were used to test the association of both knowledge and attitude levels with the different independent variables such as gender, age groups, level of training, specialty, and years of experience. For quantitative data, student independent *t*-test, paired *t*-test, and analysis of variance were used to test the association of both knowledge and attitude mean scores with these independent variables. To identify the significant predictors of physicians’ knowledge and attitudes scores on OSA and the role of dentists in OSA, multiple linear regression analysis was applied, with gender, age group, training level, specialty, years of experience, and knowledge score, as the independent variables. Significance was considered at a *p*-value < 0.05.

### 2.6. Ethical Considerations

Participation in this study was voluntary. Physicians were assured by informed consent that their responses would remain anonymous. They were asked to respond to the survey if they agreed to the informed consent. The participant’s privacy and confidentiality were assured, no identifiers were collected, and all data were kept in a secure place within Ministry of National Guard-Health Affairs (MNG-HA) premises. The study protocol (Ref. # RSS21R/014/07) received ethical approval from the Institutional Review Board (IRB) of the Ministry of National Guard-Health Affairs (MNG-HA), Riyadh, Saudi Arabia.

## 3. Results

### 3.1. Participant Demographic Characteristics

A total of 358 responses were collected (198, 55.3% males & 160, 44.7% females), after excluding responses from dentists and other non-targeted specialties. About two-thirds of the participants (223, 62.3%) were in the age group of 26–35 years. Majority of the participants were residents (146, 40.8%) and specialists (114, 31.8%). Nearly, two-thirds (226, 63.1%) of the participants were physicians practicing in general practice, family medicine, or internal medicine, and one-half (183, 51.3%) had five or more years of experience (Table 1).

### 3.2. Knowledge Assessment

Table 2 shows the knowledge statements with the frequency and percentage of responses to each statement. There was variability of responses regarding physicians’ general knowledge on OSA contributing factors and management, with 197 (55%) participants correctly reporting that a tongue with a higher fat percentage is a contributing factor for OSA, while 281 participants (78.5%) correctly reported that continuous positive airway pressure is method of management of OSA. The PMS of general knowledge was 65.6 ± 25.0 (average knowledge). Meanwhile, there was variability of responses to specific knowledge on OSA, with only 112 (12.6%) participants who disagreed that oral appliances to advance the lower jaw (mandible) during sleep for OSA management are best delivered and adjusted by physicians only, while 142 (39.7%) participants and 163 (45.5%) participants agreed that dentists could provide oral appliances for lower jaw advancement and expansion of narrowed upper jaw, respectively. The PMS of specific knowledge was 49.3 ± 19.2 (poor knowledge). 

Table 3 shows the knowledge of physicians regarding OSA and the role of dentists in its management according to some personal characteristics. The percentage mean score (PMS) of overall knowledge of all participants was 56 ± 19.4%, which is considered an “average” knowledge level. Almost half (169, 47.2%) of the participants had an average level of knowledge, one-third (127, 35%) showed poor level, while only 62 (17.3%) participants had good level of knowledge. PMS for general knowledge of craniofacial factors that may cause obstructive breathing disorders and general OSA management was 65.6 (±25.0) as compared to PMS of only 49.3 (±19.2) for the specific knowledge about the role of different dental specialties in the management of OSA (*t* = 15.23, *p* < 0.001). Meanwhile, nearly one-third of participants (127, 35.5%) reported good level of general knowledge and only 21 (5.9%) participants reported a good level of specific knowledge (χ^2^ = 143.00, *p* < 0.001). 

No sex difference was detected with regards to the level of knowledge (χ^2^ = 2.18, *p* = 0.34) or the PMS (*t* = 0.12, *p* = 0.91). As for age, PMS of knowledge was significantly higher among those aged 35 or more years (*t* = 2.31, *p* = 0.025). The proportion of those with good knowledge level increased, but not significantly, from 14.3% among younger participants to 22.2% among older participants (χ^2^= 4.19, *p* = 0.12) (Table 3).

As for the level of training, specialists had higher PMS than those of GPs (*p* < 0.001) and residents (*p* = 0.002); and residents had higher PMS than those of GPs (*p* = 0.021). Moreover, the proportion of participants with good knowledge increased significantly from 9.2% (*n* = 9) among GPs to 15.1% (*n* = 22) among residents, to 27.2% (*n* = 31) among specialists (χ^2^_LT_ = 21.88, *p* < 0.001) (Table 3).

As for specialty, there was significant difference between the different specialties in regard to the knowledge level (χ^2^ = 14.88, *p* = 0.021), with the highest proportion of good knowledge among ENT specialists (11, 33.3%), followed by pulmonologist and sleep specialists (8, 22.9%), and pediatricians (13, 20.3%), while physicians of other specialties came last (13.3%). PMS of knowledge was significantly higher for ENT specialists than among GP, FM, Int. Med. Physicians (64.1 versus 54.7, *p* = 0.009) (Table 3). 

As for years of experience, those with 5 or more years of experience, in comparison those with less than 5 years, had higher proportion of those with good knowledge (41, 22.4% versus 12.1%, χ^2^ = 8.72, *p* < 0.001) and higher PMS (59.7 versus 52.1, *t* = 3.76, *p* < 0.001) (Table 3). 

### 3.3. Attitude Assessment 

Table 4 shows the attitude statements with the frequency and percentage of responses to each statement. The statement that had the most positive and proper attitude was A1 “I believe I should pay attention to mouth breathing in patients with obstructive breathing disorders”, while the least was for A11”. I believe I should refer patients diagnosed with central sleep apnea to a dentist for a more comprehensive assessment”.

Table 5 shows the attitude of physicians towards the role of dentists in management of OSA according to some personal characteristics. PMS of attitude of all participants was 64.4 ± 17.5%, which is considered a “neutral” attitude level. About one-half (175, 48.9%) of all participants had a “neutral” attitude while only one-fourth (99, 27.6%) had a positive attitude. As for the level of training, there was a significant association between the levels of attitude and level of training. The proportion of those with positive attitude increased significantly from 15.3% (*n* = 15) among GPs to 32.2% (*n* = 47) and 32.5% (*n* = 37) among residents and specialists, respectively (χ^2^_LT_ = 11.56, *p* = 0.001). Moreover, a significant association was detected between PMS of attitude and level of training (f = 9.44, *p* < 0.001). GPs had significantly lower PMS of attitude than both residents (*p* = 0.003) and specialists (*p* = 0.001). 

As for specialty, there was a significant association between the levels of attitude and specialty (χ^2^ = 24.92, *p* < 0.001). Moreover, a significant association was detected between PMS of attitude and specialty (f = 6.29, *p* < 0.001). Pulmonology and sleep specialists had significantly lower PMS of attitude than the pediatricians (*p* < 0.001), ENT specialists (*p* = 0.002) and GP, FM, and Int. Med. Specialists (*p* = 0.017). Pediatricians had higher PMS than GP, FM, and Int. Med. Specialists (*p* = 0.007). There was no statistically significant association between the level of attitude and gender (χ^2^ = 0.15, *p* = 0.93), age (χ^2^ = 5.03, *p* = 0.08), or years of experience (χ^2^ = 1.79, *p* = 0.41)). None of these variables showed any significant association with the PMS of attitude (*p* > 0.05 each) (Table 5). 

### 3.4. Predictors of Knowledge and Attitude

Table 6 shows the multiple regression analysis of knowledge and attitude scores on OSA and role of dentists, measured by personal characteristics. It shows that the level of training was a significant predictor of higher knowledge (*t* = 3.60, *p* < 0.001) and attitude (*t* = 3.15, *p* = 0.002) scores. The knowledge score was a significant predictor of attitude score (*t* = 5.71, *p* < 0.001). 

Figure 1 shows that as the level of knowledge changed from poor to average and good levels, there was significant increase in the proportion of participants with positive attitude from 14.2% for participants with poor knowledge to 29.6% and 50% for those with average and good knowledge levels, respectively (χ^2^_LT_ = 38.10, *p* < 0.001). Moreover, PMS of attitude increased significantly from 56.4% for those with poor knowledge, to 66.8% and 74.1% for those with average and good knowledge levels, respectively (f = 27.89, *p* < 0.001). 

Figure 2 shows the association between the level of training and proportion of participants with good knowledge and positive attitude. It shows that as the level of training changed from a GP to a resident and a specialist, the proportion of participants with good knowledge increased significantly from 9.2% for GPs to 15.1% and 27.2% for residents and specialists, respectively (χ^2^_LT_ = 12.19, *p* < 0.001). Likewise, there was a significant increase in the proportion of positive attitude from 15.3% for GPs to 32.2% and 32.5% for residents and specialists, respectively (χ^2^_LT_ = 11.56, *p* < 0.001).

## 4. Discussion

Although there have been recently noticeable efforts to study and improve the awareness of dentists about OSA and its management, the efforts to evaluate physicians’ awareness and attitude towards dentists’ comprehensive role in OSA management are relatively negligible. This study was conducted as an attempt to help fill this gap. Several studies reported significant inadequate knowledge among health care professionals including dentists about OSA and its management in general [20,21,22]. In the present study, almost half (47%) of the participants had an average level of knowledge, one-third (35%) showed poor level, while only 17.3% had good level of knowledge, with a percentage mean score (PMS) of an overall knowledge of 56 ± 19.4%, which is considered an “average” knowledge level.

Dentists can have significant role in OSA management with variety of treatment modalities depending on the case: maxillary expansion, maxillary protraction, maxillomandibular surgical advancement, and mandibular advancement oral appliances [12,13,14,15]. However, physicians may lack enough awareness about the role of dentists in OSA management. The present findings showed significantly better level of general knowledge in craniofacial contributing factors and general OSA management (average level) compared to the poor level of knowledge on the specific dentists’ role in OSA management, a finding that was expected. To illustrate, PMS for general knowledge of craniofacial factors that may cause obstructive breathing disorders and general OSA management was 65.6 ± 25.0 as compared to PMS of only 49.3 ± 19.2 for the specific knowledge about the role of different dental specialties in the management of OSA (*t* = 15.23, *p* < 0.001). Nearly one-third of participants (35.5%) reported a good level of general knowledge compared to only 5.9% who reported good level of specific knowledge (χ^2^ = 143.00, *p* < 0.001). This emphasizes the fact that more efforts are needed to improve physicians’ awareness about the dentists’ role in OSA. These findings were in agreement with the findings of Sri Meenakshi et al. that the awareness of medical and dental practitioners regarding OSA diagnostic options, management, and consequences of untreated OSA was insufficient. Meanwhile, the knowledge of both groups about orthodontists’ role was poor and referral to orthodontics for OSA management was consequently rare [18]. Interestingly, the present study investigated the knowledge about the role of various dental specialists more comprehensively and targeted physicians only.

Regarding the most common craniofacial contributing factors of OSA, Sri Meenakshi et al. reported that the medical practitioners selected adenotonsillar hypertrophy as the most common one (almost 50%) [17], compared to 78.2% in the present study. The reduced jaw size had about 10% [17], as compared to 61.5% in the present study, and less than 10% selected macroglossia [17], as compared to 55% in the present study. The questionnaire of the present study included more items covering the craniofacial contributing factors of OSA, such as posteriorly placed tongue (73.5%) and retrognathia (63.1%).

It is important to mention that patient selection is without a doubt the key for successful management and that OSA diagnosis should be determined by a sleep physician. OSA related to a craniofacial deformity may be best managed by a collaboration between dental specialists (general dentists, orthodontists and/or oromaxillofacial surgeons) and physicians (primary care physicians, pediatricians, ENT specialists, pulmonologist, and sleep physicians). In the present study, about one-half (52.2%) of all participants had a “neutral” attitude while only one-fourth (24.7%) had a positive attitude towards the role of dentists in management of OSA, with a PMS of 63.1 ± 15.7%, which is considered a “neutral” attitude level. 

Sri Meenakshi et al. reported that 98% of medical practitioners surveyed were not aware of oral appliance as an option for OSA management compared to 31.6% in the present study [18]. Referral by medical practitioners to orthodontics for surgical maxillomandibular advancement or oral appliance therapy was reported by Sri Meenakshi et al. to be only 1.2% [18], which is very low compared to the findings of 71.2% and 65.4% of physicians in the present study who reported dentists could provide oral appliance therapy for OSA management and surgical jaw advancement, respectively. It is interesting that about one-half of the physicians in the present study reported that oral appliances for OSA management are best delivered and adjusted by “either physicians or dentists” (53.4%) and “physicians only” (47.6%) while only 39.7% of the physicians answered it correctly- “dentists only”. Moreover, 40.2% of physicians reported that prefabricated and custom-made oral appliances were similarly effective, however, it is well demonstrated that custom-made appliances are significantly more superior in clinical effectiveness and patient adherence [23].

A relatively small-scale survey by Jauhar et al., evaluated the attitude of 14 medical specialists and 105 general dentists towards the provision of oral appliances for snoring and sleep apnea management. They reported that all medical specialists believed that dentists had a role in managing snoring and OSA, 86% of them reported that dentists could be involved in screening, referral, and oral appliances provision, and only 57% thought that dentists could provide lifestyle advice [19]. Similarly, nearly two-thirds of physicians in the present study reported that dentists could be involved in OSA screening and referral, oral appliances provision, and lifestyle advice. The comparison with the results of Jauhar et al. may not be very accurate due to the remarkable difference in sample size [19], however, it may provide a general idea. The statement that had the least favorable attitude in the present study was A11: I believe I should refer patients diagnosed with central sleep apnea to a dentist for a more comprehensive assessment; only 34.3% selected “never” or “rarely”. It is important to distinguish between central and obstructive sleep apnea as dentists do not have a significant role in managing central sleep apnea as they do in OSA. Moreover, little more than one-third (36.3%) responded positively with “usually” or “always” to A12: I believe I should in general refer patients diagnosed with obstructive breathing disorder to dentists for a more comprehensive assessment. This is low as dentists have a significant role in OSA management. Comprehensive treatment planning and interdisciplinary management is invaluable for the most efficient patient care. 

With regards to the predictors of high knowledge and attitude scores, the present study showed that physicians’ level of knowledge was a statistically significant predictor for their attitude towards dentists’ role in OSA management. Physicians with higher knowledge scores had a more positive attitude towards dentists’ role in OSA management. This finding is logical and correlates with a previous study by Alzahrani et al. of a positive association between knowledge level and attitude of dentists towards OSA and its management [24]. Moreover, the level of training was a significant predictor of higher knowledge and attitude scores. This finding may reflect the importance of curricular as well as extracurricular dental training for physicians on OSA, and the role of dentists in its management.

### Limitations

This study may act as a pilot study to other ones from similar countries. It has some limitations. First, it relied upon a self-reported questionnaire, thus liable to recall bias, as some of the participants may have answered more favorably regarding their examination rituals in order to appear more comprehensive in their examination than what they actually carry out in practice. Second, the study is subjected to non-responder’s bias, as some of the non-responders might have had different answers than those of the responders and this might have had an implication for the generalizability of the study findings. Third, the cause-and-effect relationship between the predictors of knowledge and attitude (as an exposure) and the levels of knowledge and attitude (as an outcome), because of its cross-sectional design. Lastly, the survey was conducted via the internet, which could result in selection bias, especially that the sample was over-representative of well-educated people and those who have access to computers and the internet. Hence, it may not truly represent the entire population of the study region. Therefore, the generalization of the findings may suffer from reporting bias.

## 5. Conclusions

Physicians showed insufficient knowledge about OSA and experience a less than favorable attitude towards the role of dentists in its management. Thus, efforts to improve the study curriculum and inclusion of specialized courses about the role of dentists in OSA management are recommended, especially for general practitioners, given the serious impact and health consequences of OSA. Moreover, we encouraged there to be more comprehensive clinical protocols and guidelines for OSA management considering the dentists’ role. Effective cooperation between physicians and dentists could be established through the inclusion of dental training courses in physicians’ residency curriculum and through inter-professional seminars and interaction.

## Figures and Tables

**Figure 1 ijerph-19-16126-f001:**
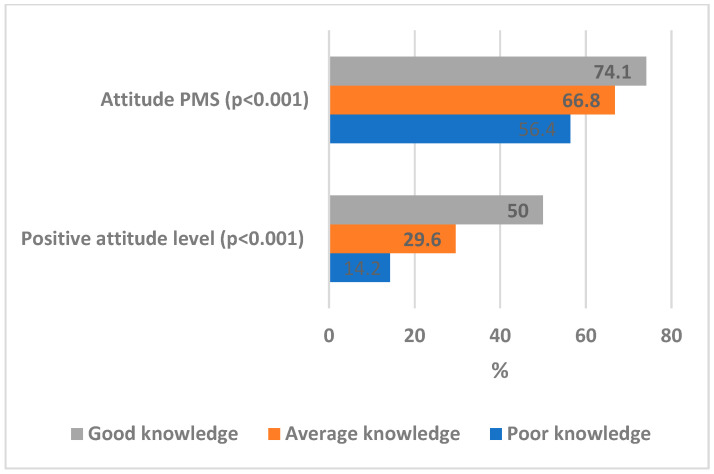
Association between the level of knowledge and attitude on OSA and the role of dentists.

**Figure 2 ijerph-19-16126-f002:**
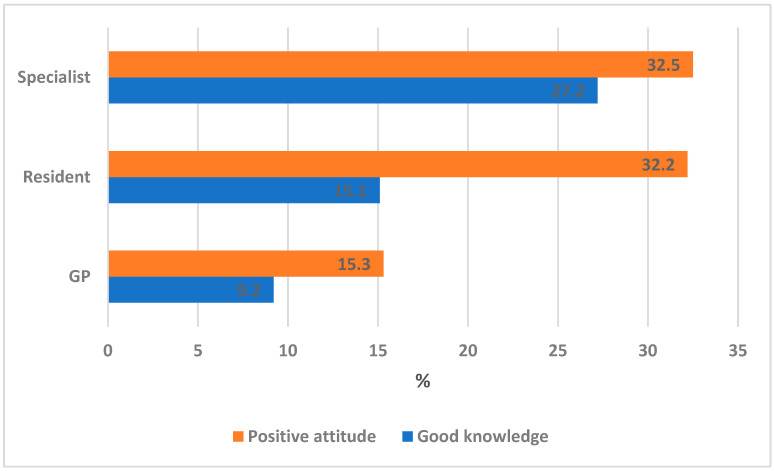
Association between the level of training and the proportion of participants with good knowledge and positive attitude.

**Table 1 ijerph-19-16126-t001:** Personal characteristics of physicians (*n* = 358).

Characteristics	No. (%)
Gender	
Male	198 (55.3)
Female	160 (44.7)
Age group (years)	
26–35	223 (62.3)
>35	135 (37.7)
Level of training	
GPs	98 (27.4)
Residents	146 (40.8)
Specialists	114 (31.8)
Specialty	
GP, FM, Int. Med.	226 (63.1)
Pediatr	64 (17.9)
Pulm./Sleep Med.	35 (9.8)
ENT	33 (9.2)
Years of experience	
<5 years	174 (48.7)
≥5 years	183 (51.3)

GPs—general practitioners, FM—Family Medicine, Int. Med.—Internal Medicine, Pediatr—Pediatrics, Pulm.—Pulmonology, ENT—Ear, Nose and Throat.

**Table 2 ijerph-19-16126-t002:** Physicians’ responses to knowledge statements on OSA.

Statements	Responses		
	True	False	Don’t Know
General Knowledge	No. (%)	No. (%)	No. (%)
Mouth breathing may be associated with:			
Enlarged adenoid	280 (78.2) *	34 (9.5)	44 (12.3)
2. Long face (increased lower facial height)	197 (55.0) *	87 (24.3)	74 (20.7)
3. Anterior open bite	215 (60.1) *	44 (12.3)	99 (27.7)
The following factors may contribute to development of OSA:			
4. Posteriorly positioned tongue	263 (73.5) *	28 (7.8)	67 (18.7)
5. Tongue with higher fat percentage	197 (55.0) *	71 (19.8)	90 (25.1)
6. Small jaw (micrognathia)	220 (61.5) *	59 (16.5)	79 (22.1)
7. Posteriorly positioned jaw (Retrognathia)	226 (63.1) *	45 (12.6)	87 (24.3)
The following is considered one of the management methods of OSA:			
8. Continuous positive airway pressure	281 (78.5) *	43 (12)	34 (9.5)
9. Uvulopalatopharyngoplasty (UPPP)	242 (67.6) *	38 (10.6)	78 (21.8)
10. Jaw (orthognathic) surgery to advance the jaws	243 (67.9) *	44 (12.3)	71 (19.8)
11. Medications	208 (58.1) *	102 (28.5)	48 (13.4)
12. Oral appliance to advance the lower jaw (mandible) during sleep	245 (68.4) *	45 (12.6)	68 (19.0)
**Specific Knowledge**			
13. Dentists may have a role in helping patients with nasal obstruction or obstructive sleep apnea	229 (64.0) *	58 (16.2)	71 (19.8)
14. Upper jaw (maxillary) expansion may help in improving bed wetting (nocturnal enuresis) in some children	173 (48.3) *	49 (13.7)	136 (38.0)
15. Narrow upper jaw (constricted maxilla) may be presented intra-orally as a posterior cross-bite	163 (45.5) *	38 (10.6)	157 (43.9)
16. Upper jaw (maxillary) expansion with distraction osteogenesis may help in improving nasal breathing in patients with breathing problems aggravated by upper jaw narrowing	176 (49.2) *	30 (8.4)	152 (42.4)
Oral appliances to advance the lower jaw (mandible) during sleep for OSA management are best delivered and adjusted by:			
17. Physicians only	170 (68.4)	112 (12.6) *	76 (19.0)
18. Dentists only	142 (39.7) *	140 (39.1)	76 (21.2)
19. Either physicians or dentists	191 (53.4)	65 (18.2) *	102 (28.5)
Recommended oral appliance for OSA management should be:			
20. Prefabricated	122 (34.1)	94 (26.3) *	142 (39.7)
21. Custom-made	197 (55) *	63 (17.6)	98 (27.4)
22. Both are similarly effective	144 (40.2)	69 (19.3) *	145 (40.5)
Dentists could provide the following:			
23. Screening for OSA and referral to medical specialists as needed	243 (67.9) *	53 (14.8)	62 (17.3)
24. Diagnosis of OSA	161 (45.0)	136 (38.0) *	61 (17.0)
25. Lifestyle advice for OSA	224 (62.6) *	77 (21.5)	57 (15.9)
26. Expansion of narrowed upper jaw	251 (70.1) *	45 (12.6)	62 (17.3)
27. Orthopedic protraction (advancement) of deficient upper jaw	235 (65.6) *	48 (13.4)	75 (20.9)
28. Oral appliances for lower jaw advancement	255 (71.2) *	36 (10.1)	67 (18.7)
29. Surgical jaw advancement	234 (65.4) *	53 (14.8)	71 (19.8)

*—correct answer.

**Table 3 ijerph-19-16126-t003:** Levels of knowledge of physicians on OSA and the role of dentists in management according to some personal characteristics.

Characteristics	Level of Knowledge			PMS (SD)
	Poor No. (%)	Average No. (%)	Good No. (%)	
Overall knowledge (*n* = 358)	127 (35.5)	169 (47.2)	62 (17.3)	56.0 (19.4)
General knowledge	74 (20.6)	157 (43.9)	127 (35.5)	65.6 (25.0)
Specific knowledge	167 (46.6)	170 (47.5)	21 (5.9)	49.3 (19.2)
	χ^2#^ = 143.00, *p* < 0.001			*t*^@^ = 15.23, *p* < 0.001 *
Gender				
Male	71 (35.9)	88 (44.4)	39 (19.7)	55.9 (20.1)
Female	56 (35.0)	81 (50.6)	23 (14.4)	56.1 (18.5)
	χ^2^ _=_ 2.18, *p* = 0.34			*t*^#^ = 0.12, *p* = 0.91
Age group (years)				
26–35	85 (38.1)	106 (47.5)	32 (14.3)	54.2 (19.4)
>35	42 (31.1)	63 (46.7)	30 (22.2)	59.0 (19.1)
	χ^2^ = 4.19, *p* = 0.12			*t*^#^ = 2.31, *p* = 0.025 *
Level of training				
GPs	49 (50.0)	40 (40.8)	9 (9.2)	49.6 (19.3)
Residents	52 (35.6)	72 (49.3)	22 (15.1)	55.2 (19.0) *
Specialists	26 (22.8)	57 (50.0)	31 (27.2)	62.5 (18.0) *
	χ^2^_LT_ = 21.88, *p* < 0.001 *			f = 12.76, *p* < 0.001 *
Specialty				
GP, FM, Int. Med.	86 (38.1)	110 (48.7)	30 (13.3)	54.7 (19.2)
Pediatr	19 (29.7)	32 (50.0)	13 (20.3)	57.1 (18.9)
Pulm./Sleep Med.	19 (29.7)	11 (31.4)	8 (22.9)	55.1 (20.2)
ENT	6 (18.2)	16 (48.5)	11 (33.3)	64.1 (19.7) *
	χ^2^ = 14.88, *p* = 0.021 *			f = 2.37, 0.071
Years of experience				
<5 years	72 (41.4)	81 (46.6)	21 (12.1)	52.1 (19.6)
≥5 years	55 (30.1)	87 (47.5)	41 (22.4)	59.7 (18.5)
	χ^2^ = 8.72, *p* < 0.001 *			*t*^#^ = 3.76, *p* < 0.001 *

GPs—general practitioners, FM—Family Medicine, Int. Med.—Internal Medicine, Pediatr—Pediatrics, Pulm.—Pulmonology, ENT—Ear, Nose and Throat, PMS—percentage mean score, *t*^@^—paired *t*-test, *t*^#^—unpaired *t*-test, χ^2^—Pearson’s Chi-squared test, χ^2^_LT_—Chi-squared test for linear trends, χ^2^#—McNemar’s test, f—analysis of variance, *—statistically significant.

**Table 4 ijerph-19-16126-t004:** Physicians’ responses to attitude statements on OSA.

Statements	Responses				
	Never No. (%)	Rarely No. (%)	Sometime No. (%)	Usually No. (%)	Always No. (%)
I believe I should pay attention to the following in patients with obstructive breathing disorders:					
1. Mouth breathing	47 (13.1)	32 (8.9)	90 (25.1)	63 (17.6)	126 (35.2)
2. Anterior open bite	57 (15.9)	46 (12.8)	99 (27.7)	75 (20.9)	81 (22.6)
3. Narrow upper jaw (constricted maxilla)	50 (14.0)	51 (14.2)	106 (29.6)	72 (20.1)	79 (22.1)
4. Long face (dolicofacial growth pattern)	51 (14.2)	56 (15.6)	106 (29.6)	80 (22.3)	65 (18.2)
5. Enlarged tonsils	55 (15.4)	43 (12.0)	84 (23.5)	90 (25.1)	86 (24.0)
6. Teeth grinding during sleep (bruxism)	52 (14.5)	63 (17.6)	106 (29.6)	80 (22.3)	57 (15.9)
I believe I should refer patients with obstructive breathing disorder to a dentist for a more comprehensive assessment when I notice the following:					
7. Constricted maxilla or posterior cross-bite	45 (12.6)	48 (13.4)	113 (31.6)	74 (20.7)	78 (21.8)
8. Nasal obstruction with long face, open bite, & mouth breathing	56 (15.6)	65 (18.2)	98 (27.4)	80 (22.3)	59 (16.5)
9. Posteriorly placed or small jaw/s	41 (11.5%)	50 (14.0)	110 (30.7%)	87 (24.3%)	70 (19.6%)
10. Grinding during sleep (bruxism)	48 (13.4)	51 (14.2)	104 (29.1%)	87 (24.3%)	68 (19%)
11. I believe I should refer patients diagnosed with central sleep apnea to a dentist for a more comprehensive assessment ^@^	67 (18.7)	56 (15.6)	139 (38.8%)	65 (18.2%)	31 (8.7%)
12. I believe I should in general refer patients diagnosed with obstructive breathing disorder to dentists for a more comprehensive assessment	30 (8.4)	55 (15.4)	143 (39.9)	84 (23.5)	46 (12.8)

^@^—negative statement.

**Table 5 ijerph-19-16126-t005:** Attitude of physicians towards the role of dentists in OSA management according to some personal characteristics.

Characteristics	Level of Attitude			PMS (SD)
	Negative No. (%)	Neutral No. (%)	Positive No. (%)	
ALL	84 (23.5)	175 (48.9)	99 (27.6)	64.4 (17.5)
Gender				
Male (*n* = 198)	47 (23.7)	95 (48)	56 (28.3)	64.3 (18.2)
Female (*n* = 160)	37 (23.1)	80 (50)	43 (26.9)	64.5 (16.7)
	χ^2^ = 0.15, *p* = 0.93			*t* = 0.13, *p* = 0.90
Age group (years)				
26–35 (*n* = 223)	46 (20.6)	119 (53.4)	58 (26)	64.4 (18.2)
>35 (*n* = 135)	38 (28.1)	56 (41.5)	41 (30.4)	64.8 (17.1)
	χ^2^ = 5.03, *p* = 0.0.08			*t* = 0.05, *p* = 0.96
Level of training				
GPs (*n* = 98)	34 (34.7)	49 (50.0)	15 (15.3)	58.1 (17.2)
Residents (*n* = 146)	30 (20.5)	69 (47.3)	47 (32.2)	66.0 (16.8) *
Specialists (*n* = 114)	20 (17.5)	57 (50)	37 (32.5)	66.7 (17.4) *
	χ^2^_LT_ = 11.56, *p* = 001 *			f = 9.44, *p* < 0.001 *
Specialty				
GP, FM, Int. Med (*n* = 226)	51 (22.6)	123 (54.4)	52 (23)	63.3 (16.3) *
Pediatr (*n* = 64)	9 (14.1)	28 (43.8)	27 (42.2)	70.3 (17.1) *
Pulm./Sleep Med. (*n* = 35)	16 (45.7)	13 (37.1)	6 (17.1)	56.2 (19.0)
ENT (*n* = 33)	8 (24.2)	11 (33.3)	14 (42.4)	68.9 (20.3) *
	χ^2^ = 24.92, *p* < 0.001 *			f = 6.29, *p* < 0.001 *
Years of experience				
<5 years (*n* = 174)	39 (22.4)	91 (52.3)	44 (25.3)	64.4 (17.9)
≥5 years (*n* = 183)	45 (24.6)	83 (45.4)	55 (30.1)	63.3 (17.0)
	χ^2^ = 1.79, *p* = 0.41			*t* = 1.14, *p* = 0.26

GPs—general practitioners, FM—Family Medicine, Int. Med.—Internal Medicine, Pediatr—Pediatrics, Pulm.—Pulmonology, ENT—Ear, Nose and Throat, PMS—percentage mean score, *t*—student *t*-test, χ^2^— Pearson’s Chi-squared test, χ^2^_LT_— Chi-squared test for linear trend, f—analysis of variance, *—statistically significant.

**Table 6 ijerph-19-16126-t006:** Multiple regression analysis of knowledge and attitude scores on OSA by some characteristics among physicians.

	Knowledge Score			Attitude Score		
	B (SE)	*t*-Value	*p*-Value	B (SE)	*t*-Value	*p*-Value
Gender	−0.70(2.0)	−0.35	0.73	−0.06(1.78)	0.03	0.98
Age group	−1.11(2.66)	−0.42	0.68	−2.34(2. 34)	−1.01	0.32
Training level	5.35 (1.48)	3.60	<0.001*	4.16(1.32)	3.15	0.002 *
Specialty	1.11(1.04)	1.07	0.28	−0.16(0.91)	−0.18	0.86
Years of experience	3.97(2.73)	1.46	0.15	−1.06 (2.40)	0.44	0.66
Knowledge (score)				0.21 (0.04)	5.71	<0.001 *
Constant	41.97(3.24)	12.94	<0.001	44.03(3.38)	13.3	<0.001

B—coefficient of determination, SE—standard error, *t*—student *t*-test, *—statistical significance.

## Data Availability

Most of the data supporting our findings is contained within the manuscript, and all others, excluding identifying/confidential patient data, will be shared upon request.

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
