# Peer review of "Awareness and Attitude of Physicians on the Role of Dentists in the Management of Obstructive Sleep Apnea"

_ijerph, 2022, doi:10.3390/ijerph192316126_

Round 1

Reviewer 1 Report

In the abstract it's not clear the instrument to collect data.

Introduction

About prevalence the most important reference is

Heinzer R, Vat S, Marques-Vidal P, Marti-Soler H, Andries D, Tobback N, Mooser V, Preisig M, Malhotra A, Waeber G, Vollenweider P, Tafti M, Haba-Rubio J. Prevalence of sleep-disordered breathing in the general population: the HypnoLaus study. Lancet Respir Med. 2015 Apr;3(4):310-8. doi: 10.1016/S2213-2600(15)00043-0. Epub 2015 Feb 12. PMID: 25682233; PMCID: PMC4404207.

About the interaction between BMI and craniofacial morphology the most recent reference is

Bertuzzi F, Santagostini A, Pollis M, Meola F, Segù M. The Interaction of Craniofacial Morphology and Body Mass Index in Obstructive Sleep Apnea. Dent J (Basel). 2022 Jul 19;10(7):136. doi: 10.3390/dj10070136. PMID: 35877410; PMCID: PMC9317640.

Please explain how did you construct the questionnaire.

Reference 15 is incomplete. 

Author Response

Reviewer # 1

Comments and Suggestions for Authors

In the abstract, it's not clear the instrument to collect data.

Agree. More description of the tool used was added in the abstract.

Introduction

About prevalence the most important reference is

Heinzer R, Vat S, Marques-Vidal P, Marti-Soler H, Andries D, Tobback N, Mooser V, Preisig M, Malhotra A, Waeber G, Vollenweider P, Tafti M, Haba-Rubio J. Prevalence of sleep-disordered breathing in the general population: the HypnoLaus study. Lancet Respir Med. 2015 Apr;3(4):310-8. doi: 10.1016/S2213-2600(15)00043-0. Epub 2015 Feb 12. PMID: 25682233; PMCID: PMC4404207.

That is a good point, a  reference was added to the manuscript. [Ref.#3]

About the interaction between BMI and craniofacial morphology the most recent reference is

Bertuzzi F, Santagostini A, Pollis M, Meola F, Segù M. The Interaction of Craniofacial Morphology and Body Mass Index in Obstructive Sleep Apnea. Dent J (Basel). 2022 Jul 19;10(7):136. doi: 10.3390/dj10070136. PMID: 35877410; PMCID: PMC9317640.

Good point, a  reference was added to the manuscript [Ref.#10]

Please explain how did you construct the questionnaire.

Re-explained in the methodology section under data collection methods

“A pilot study was conducted on ten subjects to test the reliability and feasibility of the questionnaire The questionnaire was assessed in terms of internal consistency. Cronbach's alpha was computed, and a coefficient alpha of 0.83 was considered adequate. Test-retest reliability was also assessed using Cronbach's alpha and Pearson's correlation coefficient (r). Construct validity of the checklist was assessed using expert opinion, and the final version was approved after making the necessary modifications.”

Reference 15 is incomplete

Corrected, with thanks.

Reviewer 2 Report

This multi-center study based on an online-survey reports on physicians knowledge and attitude towards OSA-management by dentists. I think it is an interesting concept worth mentioning, since raising OSA awareness worldwide in all specialties is important.

However, I have some concerns:

The first part of the introduction is quite extensive, I would shorten it, I would rather like to know: 

Who treats OSA patients in Saudi-Arabia primarily, where do patients present?

I would like to know why dentists themselves were not asked? Why were they not targeted? It would have been interesting to know how important they estimate themselves and if they even feel qualified to treat OSA. At least in Europe, not every dentist or maxillofacial surgeon is skilled /has the knowledge to treat OSA. 

I was a little confused by the questionnaire? Was the questionnaire validated? Where did the questions come from, where they taken from another study?

I must mention this: Is dentists'  treatment of OSA really the point?

It should be pointed out that a dentists’ treatment can help in some of OSA cases but not always. The goldstandard is still PAP-therapy, so practitioners who deal and manage this kind of therapy are the first ones to target if OSA is suspected. I think this should be pointed out more clearly in the introduction AND the discussion part  in order to raise OSA awareness among the readers. 

The conclusion should not be, that every OSA suspect needs to see the dentist.

Statistics are solid

Lines 309 following was the questionnaire identical to Sri Meenakshi? This is not clear. 

324 are the two studies really that comparable, the awareness is somehow biased since almost the whole questionnaire is about dentistry.

I myself as a sleep specialist totally disagree with A12. The dentists or maxillofacial surgeon maybe useful in selected cases, but I would have answered “rarely” to that question. So what can you really draw from that question?

Is it really a positive attitude if I attribute to dentists characteristics and skills (such as the treatment of central sleep apnea) that they actually cannot fulfill? doesn't it actually show ignorance?

Author Response

Reviewer # 2

Comments and Suggestions for Authors

This multi-center study based on an online-survey reports on physicians’ knowledge and attitude towards OSA-management by dentists. I think it is an interesting concept worth mentioning, since raising OSA awareness worldwide in all specialties is important.

However, I have some concerns:

The first part of the introduction is quite extensive, I would shorten it, I would rather like to know: Who treats OSA patients in Saudi-Arabia primarily, where do patients present?

I would like to know why dentists themselves were not asked? Why were they not targeted? It would have been interesting to know how important they estimate themselves and if they even feel qualified to treat OSA. At least in Europe, not every dentist or maxillofacial surgeon is skilled /has the knowledge to treat OSA.

 Good point. It was explained in the introduction and discussion that there have already been several previous studies on dentists' knowledge of OSA. However, physicians' knowledge about dentists’ comprehensive role in OSA management has not been studied as much, especially in Saudi Arabia. Dentists treating OSA in Saudi Arabia have additional training in dental sleep medicine.

I was a little confused by the questionnaire? Was the questionnaire validated? Where did the questions come from, where they taken from another study?

This was re-explained in the methodology section under data collection methods

“Based on the literature,18,19 with some modifications, an e-questionnaire was created and validated by the expert opinion. A pilot study was conducted on ten subjects to test the reliability and feasibility of the questionnaire The questionnaire was assessed in terms of internal consistency. Cronbach's alpha was computed, and a coefficient alpha of 0.83 was considered adequate. Test-retest reliability was also assessed using Cronbach's alpha and Pearson's correlation coefficient (r). Construct validity of the checklist was assessed using expert opinion, and the final version was approved after making the necessary modifications.”

I must mention this: Is dentists' treatment of OSA really the point?

It should be pointed out that a dentists’ treatment can help in some of OSA cases but not always. The gold standard is still PAP-therapy, so practitioners who deal and manage this kind of therapy are the first ones to target if OSA is suspected. I think this should be pointed out more clearly in the introduction AND the discussion part in order to raise OSA awareness among the readers. The conclusion should not be, that every OSA suspect needs to see the dentist.

Agree. This was emphasized more clearly in the introduction and discussion, as advised.

Lines 66-68 & Lines 335-337.

Statistics are solid

Thank you.

Lines 309 following was the questionnaire identical to Sri Meenakshi? This is not clear.

Good point. The questionnaires were not identical but had some similar questions. Similar questions/items were compared only. This was re-clarified. Lines 315-317.

324 are the two studies really that comparable, the awareness is somehow biased since almost the whole questionnaire is about dentistry.

Good point. The two studies had similar questions in their questionnaires: “craniofacial contributing factors of OSA.” Similar questions/items were compared only. This was re-clarified in Lines 335-337, 342-344.

I myself as a sleep specialist totally disagree with A12. The dentists or maxillofacial surgeon maybe useful in selected cases, but I would have answered “rarely” to that question. So what can you really draw from that question?

The retrognathic mandible is one of the factors that may contribute to OSA, and maxillomandibular advancement is one of the Tx options for certain cases [Ref.#14]. Of course, the sleep physician is the one who should diagnose OSA and lead the team, and PAP remains the gold standard for OSA management.  Having a Dentist/maxillofacial surgeon/ orthodontist opinion may be helpful for a more comprehensive/ efficient management to give the patient “jaw surgery” or ‘oral appliance” as an option.

 Is it really a positive attitude if I attribute to dentists characteristics and skills (such as the treatment of central sleep apnea) that they actually cannot fulfill? doesn't it actually show ignorance

Agree with you. That was previously marked as a negative, NOT positive statement, as shown in table 5. Dentists do not have a significant role in central sleep apnea management.

I believe I should refer patients diagnosed with central sleep apnea to a dentist for a more comprehensive assessment@

@__negative statement.”   --- footnote

Meaning that the positive attitude in this statement is rarely/never

Reviewer 3 Report

The manuscript entitled "Awareness and Attitude of Physicians on the Role of Dentists 2 in the Management of Obstructive Sleep Apnea" brings some important information in the field of OSA diagnosis and management but it needs some minor corrections. Here are my comments:

Page 1, line 41-42 – the authors are presenting the results of the survey in Saudi arabia, therefore the prevalence of OSA in the USA is irrelevant. Data about the prevalence in Saudi arabia are sufficient.

Page 2, line 54 – please use only the abbreviation once it has been described, throughout the whole manuscript

Please decide which term to use dentistry or dentists and unify it throughout the whole manuscript.

Page 3, line 102-104 – how did you do the validation of the questionnaire? Which questionnaire was used?

Page 3 – in Data analysis section each test should be accompanied with specific variables and outcomes tested by it (as you stated for the regression analysis).

In the Results section it would be advantageous to have whole numbers accompanying the percentages in the text to make it more clear to the readers.

Please check the spelling in Table 3 and Table 5, it seems digit 0 was used instead of letter o

Author Response

Reviewer # 3

Comments and Suggestions for Authors

The manuscript entitled "Awareness and Attitude of Physicians on the Role of Dentists 2 in the Management of Obstructive Sleep Apnea" brings some important information in the field of OSA diagnosis and management but it needs some minor corrections. Here are my comments:

Page 1, line 41-42 – the authors are presenting the results of the survey in Saudi arabia, therefore the prevalence of OSA in the USA is irrelevant. Data about the prevalence in Saudi Arabia are sufficient.

Good point. Added also another prevalence in a European country. Line 45

Page 2, line 54 – please use only the abbreviation once it has been described, throughout the whole manuscript

Corrected with thanks

Please decide which term to use dentistry or dentists and unify it throughout the whole manuscript.

Good point. It was unified to dentists, all throughout the manuscript.

Page 3, lines 102-104 – how did you do the validation of the questionnaire? Which questionnaire was used?

This was re-explained in the methodology section under data collection methods

“Based on the literature,18,19 with some modifications, an e-questionnaire was created and validated by the expert opinion. A pilot study was conducted on ten subjects to test the reliability and feasibility of the questionnaire The questionnaire was assessed in terms of internal consistency. Cronbach's alpha was computed, and a coefficient alpha of 0.83 was considered adequate. Test-retest reliability was also assessed using Cronbach's alpha and Pearson's correlation coefficient (r). Construct validity of the checklist was assessed using expert opinion, and the final version was approved after making the necessary modifications.”

Page 3 – in Data analysis section each test should be accompanied with specific variables and outcomes tested by it (as you stated for the regression analysis).

Agree. All the exposure and outcome variables for different tests were mentioned as appropriate. [ Data analysis, p 4&5]

In the Results section it would be advantageous to have whole numbers accompanying the percentages in the text to make it more clear to the readers.

Agree. For all percentages, the corresponding numbers were inserted in the results text

Please check the spelling in Table 3 and Table 5, it seems digit 0 was used instead of letter

Corrected, with thanks.